# A Possible Preventive Role of Physically Active Lifestyle during the SARS-CoV-2 Pandemic; Might Regular Cold-Water Swimming and Exercise Reduce the Symptom Severity of COVID-19?

**DOI:** 10.3390/ijerph18137158

**Published:** 2021-07-04

**Authors:** Viktor Bielik, Marian Grendar, Martin Kolisek

**Affiliations:** 1Department of Biological and Medical Science, Faculty of Physical Education and Sport, Comenius University in Bratislava, 814 69 Bratislava, Slovakia; 2Biomedical Center Martin, Jessenius Faculty of Medicine in Martin, Comenius University in Bratislava, 036 01 Martin, Slovakia; Marian.Grendar@uniba.sk (M.G.); martin.kolisek@uniba.sk (M.K.)

**Keywords:** cold-water swimming, acute respiratory infection, athletes, physical activity, BMI

## Abstract

The objective of this study was to investigate the incidence and course of COVID-19 and the risk of an upper respiratory tract infection in a group of people with physically active lifestyles. Data were collected anonymously using an online survey platform during December 2020. The age of participants ranged from 18 to 65 years. Out of 2343 participants, 11.5% overcame COVID-19 infection. Relative to the control group (CTRL), physically active, cold-water swimmers (PACW) did not exhibit a lower risk of incidence for COVID-19 (RR 1.074, CI 95% (0.710–1.625). However, PACW had a higher chance of having an asymptomatic course of COVID-19 (RR 2.321, CI 95% (0.836–6.442); *p* < 0.05) and a higher chance of only having an acute respiratory infection once or less per year than CTRL (RR 1.923, CI 95% (1.1641–2.253); *p* < 0.01). Furthermore, PACW exhibited a lower incidence of acute respiratory infection occurring more than twice per year (RR 0.258, CI 95% (0.138–0.483); *p* < 0.01). Cold-water swimming and physical activity may not lessen the risk of COVID-19 in recreational athletes. However, a physically active lifestyle might have a positive effect on the rate of incidence of acute respiratory infection and on the severity of COVID-19 symptoms.

## 1. Introduction

Cold-water swimming involves whole-body immersion in cold-water outdoors (lake, river, sea, etc.) mainly during the winter season or in the colder regions. The taking of a cold-water bath together with winter or ice swimming has long been a tradition in northern countries [1]. Until a few years ago, cold-water swimming was practiced by a very close community of enthusiasts. Today, it is a popular sport and recreational activity around the world, with 50 countries having branches of the International Ice Swimming Association (IISA) [2]. Cold-water treatment is believed to contribute to regeneration after a period of intense exercise. During the procedure, a substantial part of the human body or the whole body is immersed in a bath of cold or ice-containing water for a limited duration. Evidence suggests that repeated cold-water therapy has a wide variety of health benefits [3,4,5] and that winter swimming has a positive impact on the immune system [1].

Similarly, a plethora of health benefits of physical activity has been documented in recent decades [6]. Indeed, a positive linear relationship seems to exist between physical activity and the health status of an individual. Thus, increased intensity of physical activity and above-average fitness lead to additional improvements in the health status of humans [7]. On the contrary, an unhealthy lifestyle, obesity, and chronic ailments jointly impair immune functions/responses and increase the risk of severe infectious disease [8].

Recently, Sallis et al. [9] reported that physically inactive patients were more likely to be hospitalized with COVID-19 and had a greater risk of admission to intensive care units and of mortality. However, still, a healthy lifestyle and physical fitness have been speculated to be key factors in resisting infection with SARS-CoV-2 [10,11]. These assumptions have been made based on reports correlating increased resistance to viruses such as SARS-CoV-2 with modification of the risk factors for obesity and chronic diseases [8,12]. Practical recommendations and exercise advice have been published during the COVID-19 pandemic in order to address the global problem of lower physical activity and to encourage active lifestyles [13,14,15,16]. Nevertheless, whether regular cold-water swimming and/or physical activity have superior benefits for the prevention of COVID-19 and/or contribution to an asymptomatic or a less severe course of the disease needs to be better understood. Furthermore, knowledge is limited concerning the annual number of infections of the upper respiratory tract in cold-water swimmers and the physically active [17].

The purpose of this study has therefore been to identify the existence of any differences between cold-water swimmers, physically active subjects, and healthy controls with regard to the incidence of COVID-19 and upper respiratory tract infections. Furthermore, we aimed to examine whether COVID-19-infected cold-water swimmers and athletes experienced milder symptoms than a sedentary control group. Our results lead us to hypothesize that a combination of cold-water swimming and regular sport has substantial preventive benefits for reducing upper respiratory tract infections and COVID-19 severity.

## 2. Materials and Methods

### 2.1. Participants

A total of 2343 participants completed our questionnaire. Participants were recruited via flyers, print and web advertisements, social media, and websites of the Slovak Ice-Bears Organization and various athletic clubs. Moreover, students of the Faculty of Physical Education and Sport, Comenius University in Bratislava were addressed due to sport history. Students of Faculty of Electrical Engineering and Information Technology, the Slovak University of Technology in Bratislava were approached with the assumption of a sedentary lifestyle. Data were collected anonymously by using the online survey platform Survio (Brno, Czech Republic). Informed consent was provided by participants ticking a mandatory box presented prior to their proceeding to the survey. Participation was voluntary, and no financial compensation or course credit was provided in exchange for participation. The data collection began on 7 December and ended on 18 December 2020. On 11 August, the Minister of Health confirmed the second wave of the pandemic [18]. The Government of the Slovak Republic declared a state of emergency on 1 October 2020, which lasted until 14 May 2021 [19,20]. As of 12 October, teaching at all secondary schools was interrupted and was switched to distance learning. On 18 December, the Statistical Office of the Slovak Republic reported 149,275 positive COVID-19 patients from 3,718,060 Slovaks (15–65 years).

### 2.2. Data Collection

The following inclusion criteria were specified to create a group: (1) physically active, cold-water swimmers (defined as frequency of 2–3 cold-water swimming sessions/week, experience with cold-water swimming ≥2 seasons, frequency of exercise ≥3 active sport sessions/week), (2) physically active (frequency of exercise ≥3 active sport sessions/week), and (3) control subjects (≤120 min physical exercise/week). An electronic questionnaire (SF1) was used to collect anonymous data. The items were piloted with a smaller representative group of cold-water swimmers (*n* = 404) to check item saliency, relevance, appropriateness, and clarity in capturing the construct of interest. All participants provided favorable responses and offered valuable feedback regarding the content and clarity of the items and instructions. Development of the questionnaire including a pilot study and the use of appropriate language resulted in a final version of the questionnaire that was understandable among the target population of recreationally active and sedentary responders and was adapted according to The International Physical Activity Questionnaire-Short Form [21]. The participants were also asked to document the frequency of their cold-water swimming sessions (number of sessions per week and years of cold-water swimming), symptoms of COVID-19 if they had tested positive (asymptomatic, mild, moderate, severe), hours of sleep per night, and the frequency of any upper respiratory tract infections (annual number of infections). All the questions could be answered in a maximum of 5 min. The feedback necessary for responding was, in general, positive. Respondents were asked to select appropriate answers from the offered options or to complete missing responses on the spot (numeric data). Body mass index (BMI) was calculated from self-reported height and weight [22]. The data collectors did not participate in the study.

COVID-19 tests were performed on samples from combined nose and throat swabs by using the real-time polymerase chain reaction (RT-PCR) in accredited laboratories. Participants who were shown to be positive for COVID with tests other than RT-PCR (e.g., the COVID-19 Antigen Test) were included separately.

### 2.3. Data Analysis

The data were explored and analyzed in R ver. 4.0.3 [23] by using the libraries beeswarm [24], randomForestSRC [25], and ggRandomForests [26] Categorical data were subjected to Pearson’s chi-squared test or the Fisher test, where appropriate. Continuous data (e.g., age) were tested by the Kruskal–Wallis test. 

The RandomForest (RF) machine-learning algorithm was trained on the data and on various subsets of the data to assess the predictive ability of the studied variables for discrimination between subjects with positive or negative findings for the COVID-19 tests. The predictive/discriminative ability of the selected predictors was assessed by the ROC curve and quantified by the area under ROC (AUC), obtained from the out-of-bag data.

Sample characteristics were summarized using frequencies and proportions. Risk ratios (RR) and 95% confidence intervals (95% CI) were reported as a means of examining the strength of association between the baseline control group and risk of positive COVID-19, symptoms of COVID-19, and incidence of upper respiratory tract infection in the various subgroups of data.

## 3. Results

The study cohort included 2343 adult community volunteers during 2020. After excluding (1) cold-water swimming newcomers (*n* = 510) and (2) participants with missing data or with off-scale data for age, sleeping hours/night, and BMI, 1544 remained for risk ratio estimation (Table 1). Of these, 26.2% (*n* = 404; male 17.6%, female 8.5% of the total probands count) were classified as physically active, cold-water swimmers, 50.2% (*n* = 775; male 24.6%, female 25.6% of the total probands count) were physically active, and 23.6% (*n* = 365; male 8.9%, female 14.8% of the total probands count) were designated as controls (Table 1).

In the original cohort (*n* = 2343), 269 (11.5%) participants had been tested positive for SARS-CoV-2 between the 1 March and the 30 November 2020, and of these, 97 (4.1%) had been tested positive for SARS-CoV-2 by the COVID-19 antigen-test, 147 (6.3%) by RT-PCR alone and 25 (1.1%) by RT-PCR and the COVID-19 antigen-test (Table 2).

### 3.1. Risk of Infection and Course of COVID-19 

In comparison with the control group, neither the physically active group nor the physically active, cold-water swimmers exhibited a lower risk of being positive for SARS-CoV-2 (RR 0.992, 95% CI, 0.685 to 1.439; RR 1.074, 95% CI 0.710 to 1.625) (Table 3). The physically active, cold-water swimmers had more than a two-fold higher chance of having an asymptomatic course of COVID-19 when they tested positive for SARS-CoV-2 (RR 2.321, CI 95% 0.836 to 6.442). Neither the physically active group nor the physically active, cold-water swimmers exhibited a lower risk of having moderate or severe symptoms of COVID-19.

### 3.2. Risk of Upper Respiratory Tract Infection

In comparison with the control group, the physically active group had a slightly lower incidence of upper respiratory tract infections occurring more than twice per year (RR 0.718, CI 95% 0.496 to 1.038) (Table 3). Similarly, the physically active, cold-water swimmers had less than a 0.3-fold lower incidence of having upper respiratory tract infections occurring more than twice per year (RR 0.258, CI 95% 0.138 to 0.483). Furthermore, the physically active, cold-water swimmers exhibited an almost two-fold higher chance of becoming infected with such an infection only once or less per year (RR 1.923, CI 95% 1.1641 to 2.253). Risks of upper respiratory tract infection separately for males and females are presented in Appendix A.

### 3.3. RF Machine-Learning Analysis of the Ability of Selected Variables/Predictors to Distinguish Individually or Jointly between COVID-19 Positive and Negative Individuals

Next, we utilized RF machine-learning algorithm aiming to identify variable(s) able to discriminate between subjects with positive or negative findings for the COVID-19 tests. The following variables were used as predictors: frequency of inflammation of upper respiratory tract, weight, height, age, gender, cold/ice water swimmer (yes/no), sportsman (yes/no), and frequency of cold/ice water swimming. RF ranked them by variable importance. No formal feature selection was performed. The predictors lead to AUC = 0.537 (Figure 1); thus, leading to the conclusion that the tested variables have only minuscule power to discriminate between the two classes of subjects.

In the following, we tested whether excluding asymptomatic COVID-19 probands and those tested only with SARS-CoV2 antigen-tests may improve the performance of the RF machine-learning predictions. In this case, the predictive power was AUC = 0.599 (Figure 2), thus only slightly better than in the previous case.

## 4. Discussion

We performed a cross-sectional study with the aim of identifying/deciphering any effects of a physically active lifestyle on the prevention of COVID-19 and/or acute respiratory infections. Preliminary data suggested that people with an unhealthy lifestyle and obesity were at a higher risk of suffering from the severe form of COVID-19 [12]. Obesity, accompanied by chronic inflammation and impaired immunity responses, was associated with several debilitating and life-threatening disorders. These include: respiratory dysfunction, metabolic syndrome, diabetes, cardiovascular diseases, and some malignancies. Certain metabolic risk factors are widely recognized as being related to the more severe form of COVID-19 [12]. Therefore, based on the latter, we considered the proposition that a physically active lifestyle (cold-water swimming and/or physical activity) might be advantageous in resisting infection by SARS-COV-2 [10,11]. Based on reported weekly hours of exercise, we assumed that the physically active responders have higher physical fitness than that of the control group. Brawner et al. [27] have investigated the impact of cardio-respiratory fitness (CRF), indicated by peak metabolic equivalents (METs), on hospitalization risk attributable to COVID-19. They have reported lower peak METs (6.7 ± 2.8 METs) in hospitalized patients compared with those not being hospitalized (8.0 ± 2.4 METs; *p* < 0.001). They conclude that CRF is independently and inversely associated with the likelihood of hospitalization because of COVID-19 [27]. This is in accordance with our study in which we report no severe COVID-19 cases with hospitalization in the group with a physically active lifestyle. However, our finding might be correlated with the nature of the cohort in our study, which consisted of relatively young individuals with low BMI. Of note, the BMI of the participants in the study of Brawner et al. (34) was on average 32.7 kg/m^2^, which corresponds with obesity class I [28], whereas the severity of COVID-19 is generally accepted to be considerably worse in obese individuals [29,30].

Surprisingly, our data showed that neither physical activity nor cold-water swimming is correlated with a lower incidence of COVID-19 when compared with the controls. Ho et al. [31] have investigated several risk factors for COVID-19. From their analysis of 235,928 eligible participants aged from 49 to 83 years, poor physical fitness (as measured by a slow walking pace), smoking, BMI, and hypertension are associated with COVID-19 [31]. On the other hand, Zbinden-Foncea [32] suggested that high levels of cardiorespiratory fitness are immuno-protective in patients who contract SARS-CoV-2. They speculate that the positive effects of moderate doses of exercise on immune protection against COVID-19 are mediated by modulation of the activity of angiotensin-converting enzyme 2. This hypothesis has subsequently been corroborated by Motta-Santos and colleagues [33] who consider that the anti-inflammatory actions of angiotensin-(1–7) represent a positive effect of physical exercise. Notably, the physically active, cold-water swimmers in our study had a more than two-fold higher chance of having the asymptomatic form of COVID-19 when they tested positive for the disease. We speculate that they are asymptomatic because of the effect of their physically active lifestyle on their anti-inflammatory adaptation and inflammatory reactions. Recent studies suggest that physical activity mitigates some features of immunosenescence and thus improves immune responses. However, the effects of immunosenescence on the immune response against SARS-CoV-2 have been little explored [34].

One mechanism explaining the higher chance of an asymptomatic course of COVID-19 related to a physically active lifestyle might be the metabolism carried out by mitochondria, which are pivotal for the immune response [35]. Many viruses modulate the functions of mitochondria and thus affect their performance and homeostasis [36]. Therefore, differential alterations of mitochondrial functions might explain, at least to some extent, the variability in responses to SARS-CoV-2 infection among patients [10]. Nunn et al. [10] have speculated that the maintenance of “mitochondrial health” is vital for SARS-CoV-2 resistance, which is promoted by an effective mitochondrial reserve induced by factors such as physical activity. Furthermore, the key player in anti-inflammatory adaptation is mitochondrial stress, which enhances mitochondrial functions not only in muscle, but in various other organs in which myokines playing a key role [37]. Irisin, a thermogenic hormone-like adipomyokine possessing mitochondria-protective functions can protect against ischemia/reperfusion injury in the lung [38]. It is produced in abundance by skeletal muscle in response to exercise. Once released into the circulation, irisin acts on white adipocytes by inducing the browning response [39]. Positive effects of irisin on the expression of multiple genes related to viral infection by SARS-CoV-2 have been reported in human subcutaneous adipocytes [40]. The activity of myokines can be modulated by adipokines, confirming that crosstalk occurs between skeletal muscle and adipose tissue during metabolic regulation [41]. Exercise-induced myokines appear to be involved in mediating both systemic and local anti-inflammatory effects [42].

Mitochondrial functions and capacity can also be enhanced by brown adipose tissue, which is metabolically activated by cold exposure [43,44]. Recently, Kovaničová et al. [45] have reported that cold-acclimatized ice-water swimmers respond to cold by non-shivering thermogenesis mediated by higher fat oxidation than controls. However, it is important to note the ambient temperature (e.g., water). Swimming in mild-cold water (20 °C) does not activate brown adipose tissue in mice [46]. Similarly, mild cold exposure in humans does not stimulate thyroid hormone circulation [45], which has been shown to induce adipose tissue browning [47]. We assumed that the novice cold-water swimmers might have a lower adaptation to cold than experienced swimmers, and therefore, more than five hundred first-season cold-water swimmers were excluded from our final analysis. However, we do not question the positive effects of acute cooling. Indeed, the efficacy of a single exposure to cold-water immersion on recovery has been established in elite professional footballers [48]. The meta-analytical review of Poppendieck [49] has further shown that the percentage improvements of performance recovery to be expected from post-exercise cooling are large enough to be relevant for competitive athletes. Moreover, the positive effects of cooling have been demonstrated on the reduction of muscle soreness and of markers of muscle damage [49].

As described above, acute exposure to cold water temperatures is beneficial to athletic performance and recovery. However, variable degrees of body adaptation to cold only appear when such exposure is repeated regularly and long enough [50]. According to the data from our study, the physically active, cold-water swimmers had an almost two-fold higher chance of having an acute respiratory infection only once or less per year. Moreover, the physically active, cold-water swimmers exhibited less than a 0.3-fold lower incidence of an upper respiratory tract infection occurring more than twice per year. Thus, our results are in accordance with previously published investigations, reporting that the annual number of infections of the upper respiratory tract in cold-water swimmers decreased by more than 40% when compared with those of controls [17]. Recently Kunutsor and Laukkanen [51] have evaluated the independent and joint associations of frequency of sauna bathing and cardiorespiratory fitness with pneumonia risk in 2275 men from 42 to 61 years of age. They have found that a combination of high fitness levels and frequent sauna baths is associated with a substantially lowered risk for future pneumonia compared with each modality alone. The implications of their findings with regard to altering COVID-19 disease or its severity should thus be further examined [51].

A physically active lifestyle is known to boost the health of humans. Such a lifestyle might positively impact (reduce) the risk of becoming infected and adversely affected by various pathogens including SARS-CoV-2 [52]. An unhealthy lifestyle (e.g., sedentary way of life, smoking, obesity) seems to be a major risk factor for COVID-19 hospital admission. The adoption of some simple lifestyle changes might lower the risk of severe infection [11]. Unfortunately, our sample does not allow detailed analysis in this regard, but the post-hoc analyses show that physically active males are at a lower rate of incidence of acute respiratory infections per year.

The main limitations of this study are that the measurements of physical activity and cold-water swimming were self-reported. Notably, because this is an observational study, we cannot clearly conclude that a physically active lifestyle with cold-water swimming is causally related to a higher probability of an asymptomatic COVID-19 course. Further limitations of our cohort are that weight was in the normal range, and that the individuals were relatively young. Therefore, we assume that the use of a more heterogeneous cohort would lead to more significant differences. Finally, we cannot distinguish the effect of swimming in cold water only because the ice-water swimmers reported additional physical activities. Our findings substantiate the need for an adequate physically active lifestyle and the need for governmental measures such as the meaningful support of sport/physical activities during pandemics and the massive promotion of a healthy diet. Future assessments of other risk and risk-reducing factors, such as the type, intensity, and length of physical exercise, time of cold-water swimming, and nutritional status will be necessary.

## 5. Conclusions

A physically active lifestyle, namely a combination of regular physical activity and cold-water swimming, potentially lowers the incidence of acute respiratory infection. However, cold-water swimming and physical activity might not lessen the risk of COVID-19 in recreational athletes. Despite individuals with an unhealthy lifestyle and obesity being at increased risk of severe COVID-19, no evidence was presented that exercise training or cold-water swimming reduces the incidence of COVID-19. Therefore, athletes are at similar risk for COVID-19 infection as control subjects.

## Figures and Tables

**Figure 1 ijerph-18-07158-f001:**
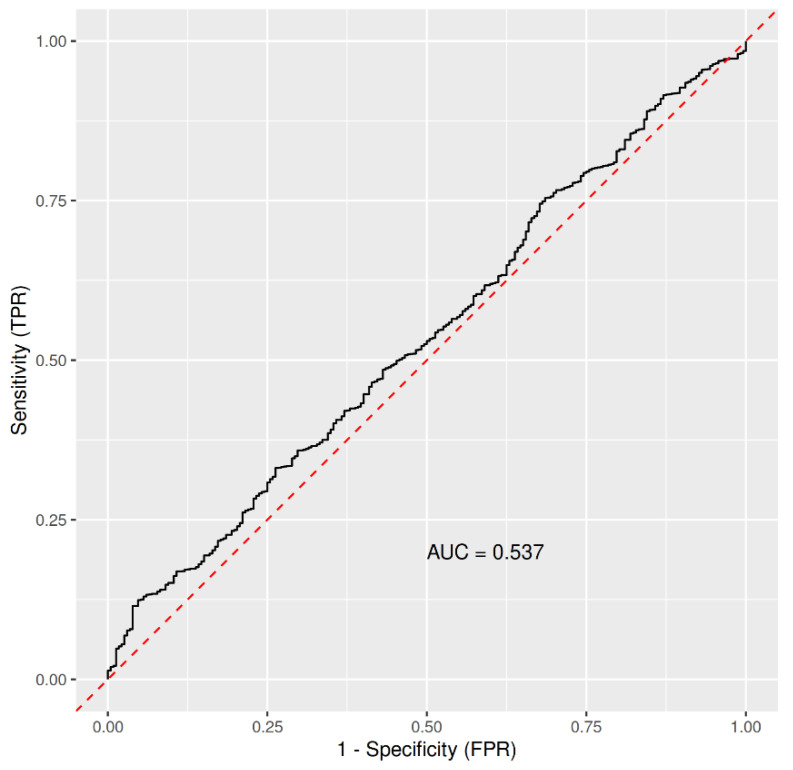
ROC based on the Out-Of-Bag data from imbalanced RandomForest, trained on 2111 controls (probands who were not infected with COVID-19) and 232 COVID-19 cases, with frequency of inflammation of upper respiratory tract, weight, height, age, gender, being a cold/ice water swimmer, being a sportsman, and frequency of cold/ice water swimming as predictors. Abbreviations: AUC, area under the curve; FPR, false positive rate; TPR, true positive rate.

**Figure 2 ijerph-18-07158-f002:**
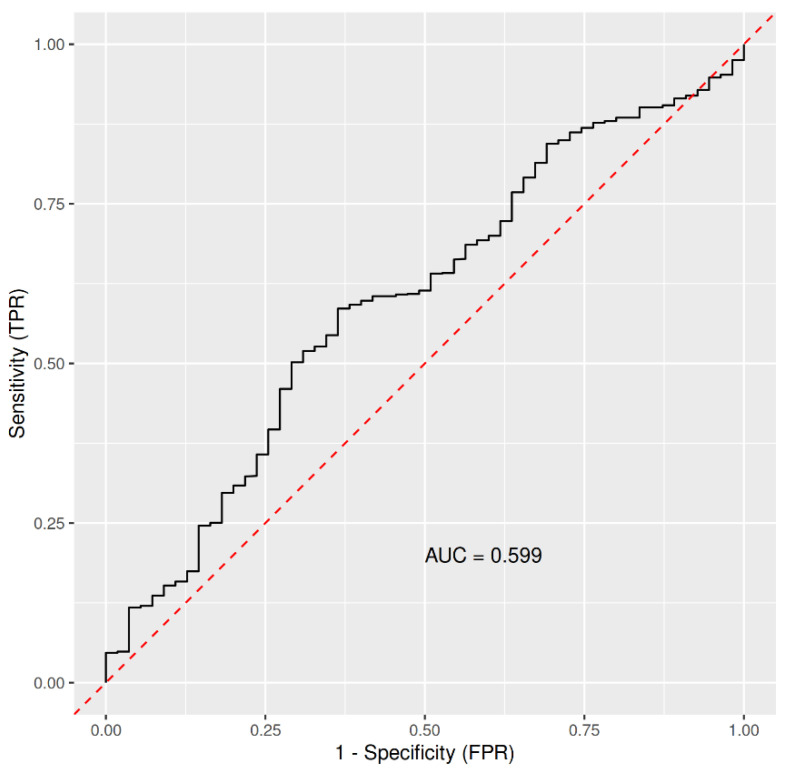
ROC based on the Out-Of-Bag data from imbalanced RandomForest, trained on 1130 controls (probands who were not infected with COVID-19) and 55 COVID-19 cases detected by RT-PCR, with the asymptomatic cases excluded, where frequency of inflammation of upper respiratory tract, weight, height, age, gender, being a cold-water swimmer, being a sportsman, and frequency of cold/ice water swimming were used as the predictors. Abbreviations: AUC, area under the curve; FPR, false positive rate; TPR, true positive rate.

**Table 1 ijerph-18-07158-t001:** Physical and lifestyle characteristics of the cohort of analyzed individuals (n=1544).

	Male *	Female *
age (years)	30.5 (29.7–31.3)	35.1 (34.4–35.8)
height (cm)	181.6 (181.0–182.0)	167.4 (167–168)
weight (kg)	83.1 (82.2–84.0)	67.0 (66.1–67.9)
BMI	25.2 (25.0 to 25.4)	23.9 (23.6–24.2)
physical activity	male (*n*)	female (*n*)
sedentary	137	228
recreational athlete (3–5 h/week)	225	295
recreational athlete (6–9 h/week)	128	90
sub-elite athlete (>10 h/week)	27	10
cold-water swimming (year)	male (*n*)	female (*n*)
none	517	623
second to third	193	102
fourth and more	79	30
* Data are presented as mean and (95 %CI)		

**Table 2 ijerph-18-07158-t002:** Data distribution with respect to diagnostic test used for detecting COVID-19 (*n* = 2343).

	Antigen Test	PCR and Antigen Test	PCR Test	*p*
gender				0.62
male	48 (49%)	12 (48%)	81 (55%)	
female	49 (51%)	13 (52%)	66 (45%)	
age (years)	31 (22, 43)	28 (22, 42)	31 (22, 42)	0.88
height (cm)	174 (167, 180)	176 (168, 183)	175 (168, 183)	0.49
weight (kg)	78 (63, 86)	73 (60, 86)	75 (62, 86)	0.76
physical activity				
sedentary	21 (22%)	5 (20%)	33 (22%)	
recreational athlete (3–5 h/week)	46 (47%)	9 (36%)	60 (41%)	
recreational athlete (6–9 h/week)	24 (25%)	8 (32%)	41 (28%)	
sub-elite athlete (>10 h/week)	6 (6.2%)	3 (12%)	13 (8.8%)	
cold-water swimming (year)				0.76
none	53 (55%)	14 (56%)	76 (52%)	
first	22 (23%)	4 (16%)	30 (20%)	
second to third	10 (10%)	3 (12%)	27 (18%)	
fourth to fifth	7 (7%)	2 (8.0%)	8 (5.4%)	
sixth and more	5 (5%)	2 (8.0%)	6 (4.1%)	
cold-water swimming (days/week)				0.65
none	50 (52%)	13 (52%)	76 (52%)	
once	13 (13%)	3 (12%)	31 (21%)	
twice	24 (25%)	7 (28%)	27 (18%)	
three times and more	10 (10%)	2 (8%)	13 (8.8%)	
positive for COVID-19				0.00
no	31 (32%)	7 (28%)	10 (6.8%)	
yes (from March to September)	5 (5.2%)	2 (8.0%)	21 (14%)	
yes (from October to December)	61 (63%)	16 (64%)	116 (79%)	
course of COVID-19				0.06
asymptomatic	22 (29%)	2 (11%)	35 (25%)	
mild	29 (39%)	10 (56%)	40 (28%)	
moderate	24 (32%)	5 (28%)	61 (43%)	
severe	0 (0%)	1 (5.6%)	5 (3.5%)	
unknown	22	7	6	
incidence of ARTI/yr				0.21
<once	55 (57%)	13 (52%)	70 (48%)	
once	26 (27%)	4 (16%)	46 (31%)	
twice	10 (10%)	4 (16%)	24 (16%)	
>twice	6 (6.2%)	4 (16%)	7 (4.8%)	

Statistics presented: *n* (%), median (IQR) Statistical test performed: chi-square test of independence, Kruskal–Wallis test, Fisher test; * ARTI—acute respiratory tract infection.

**Table 3 ijerph-18-07158-t003:** Risk ratios for diagnostics and course of COVID-19 and acute respiratory infection (*n* = 1544).

	Model 1		Model 2		Model 3	
	RR (95% CI)	*p*	RR (95% CI)	*p*	RR (95% CI)	*p*
COVID-19 diagnostics						
PCR/Positive	1.091 (0.688, 1.730)	0.36	1.163 (0.696, 1.942)	0.28	1.093 (0.802, 1.488)	0.29
PCR + Antigen/Positive	0.992 (0.685, 1.439)	0.48	1.074 (0.710, 1.625)	0.37	1.025 (0.794, 1.323)	0.43
Course of COVID-19						
asymptomatic	1.677 (0.628, 4.477)	0.15	2.321 (0.836, 6.442)	0.05	1.421 (0.779, 2.590)	0.13
mild	1.207 (0.627, 2.320)	0.29	0.987 (0.457, 2.134)	0.49	1.194 (0.766, 1.859)	0.22
moderate	0.696 (0.399, 1.213)	0.1	0.829 (0.446, 1.540)	0.28	0.760 (0.504, 1.146)	0.1
severe	-		1.364 (0.229, 8.110)	0.37	1.351 (0.226, 8.068)	0.37
Incidence of acute respiratory infection/yr						
<once	1.108 (0.936, 1.311)	0.12	1.923 (1.1641, 2.253)	0.00	1.070 (0.916, 1.249)	0.20
once	1.154 (0.971, 1.370)	0.05	0.578 (0.457, 0.731)	0.00	1.232 (0.823, 1.844)	0.16
twice	0.869 (0.680, 1.112)	0.13	0.394 (0.270, 0.574)	0.00	0.485 (0.256, 0.920)	0.01
>twice	0.718 (0.496, 1.038)	0.04	0.258 (0.138, 0.483)	0.00	0.485 (0.160, 1.476)	0.10

Model 1 physically active group; Model 2 cold-water swimmers; Model 3 males only.

## Data Availability

Not applicable.

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
