# Peer review of "A Possible Preventive Role of Physically Active Lifestyle during the SARS-CoV-2 Pandemic; Might Regular Cold-Water Swimming and Exercise Reduce the Symptom Severity of COVID-19?"

_ijerph, 2021, doi:10.3390/ijerph18137158_

Round 1

Reviewer 1 Report

The authors conducted a survey-based study to investigate the association between the lifestyle and the Covid-19 pandemic. The investigators made their best effort to control the validity of the data regarding the Covid-19 incidence. The results are concise and well presented with tables and figures. The correlation between the incidence of pulmonary infection seems to be straightforward supporting the negative association between the level of physical activity and the infection. In the discussion, the authors provided speculative mechanism that the protective role that physical activity may provide.  Although more information is needed and might be helpful to understand the demographic and social economic status of the sample cohorts in association with the infection, it sounds plausible to measure the correlation between physical activity and the infection. 

There are minor text editing needed due to extra spaces and punctuations that might need to be corrected. Without line numbers labeled, it's hard to specify. 

Author Response

There are minor text editing needed due to extra spaces and punctuations that might need to be corrected. Without line numbers labeled, it's hard to specify. 

A: Thank you for reviewing  and acknowledging the manuscript. We apologize for the typos in the text.

Reviewer 2 Report

In General: it's a good paper and the subject of the manuscript is applicable and useful. 

Title: the title properly explain the purpose and objective of the article

Abstract: abstract contains an appropriate summary for the article, language used in the abstract easy to read and understand, there are no suggestions for improvement.

Introduction: authors do provide adequate background on the topic and reason for this article and describe what the authors hoped to achieve.

Results: the results presented clearly, the authors provide accurate research results, there is sufficient evidence for each result.

Conclusion: in general: Good and the research provides sample data for the authors to make their conclusion.

Grammar: Need Some revision.

Finally, this was an appealing article, in its current state it adds much new insightful information to the field.Therefore, I accept that paper to published in your journal

Author Response

Thank you for reviewing  and acknowledging the manuscript.

Reviewer 3 Report

This study explored the potential preventive role of physical activity in the COVID-19 pandemic. The results are interesting, and the methodology is sound. I have some concern that I listed below for the authors to follow and improve their manuscript:

  • Please specify the months/days the city the study took place in was affected by local COVID restrictions/lockdown and if the data collected between December 7th and 18th was during this time.
  • Please include in a separate table the information on demographics for the whole sample without the differentiation of COVID testing.
  • It seems that the COVID symptoms severity was completed through a self-report questionnaire. How did the authors make sure participants knew the difference between different levels of COVID symptoms?
  • Did any of the demographics such as age, gender, etc.… play a moderating role in the relationships explored in the study?
  • Did the authors collect information on the type and intensity of activity performed by the physically active group? If yes, please report it.
  • It seems an important recent reference in this area is missing: “Sallis R, et al. Br J Sports Med 2021;0:1–8. doi:10.1136/bjsports-2021-104080”. This study concludes that those meeting PA guidelines consistently reported less severe COVID symptoms compared to those who were consistently inactive. I suggest adding this reference to the introduction to be able to make a stronger case for the current study’s hypothesis and the results.

Author Response

R: Please specify the months/days the city the study took place in was affected by local COVID restrictions/lockdown and if the data collected between December 7th and 18th was during this time.

A: On August 11, the Minister of Health confirmed the second wave of the pandemic. The Government of the Slovak Republic declared a state of emergency on October 1, 2020, which lasted until May 14, 2021. As of October 12, teaching at all secondary schools was interrupted and it was switched to distance learning. On 18 December, the Statistical Office of the Slovak Republic reported 149,275 positive COVID-19 patients from 3,718, 060 Slovaks (15–65 years).This information was added to Method section.

R: Please include in a separate table the information on demographics for the whole sample without the differentiation of COVID testing.

A: New table (Table 1) with information of cohort was added.

R: It seems that the COVID symptoms severity was completed through a self-report questionnaire. How did the authors make sure participants knew the difference between different levels of COVID symptoms?

A: COVID symptoms were explained in the questionnaire.

Asymptomatic (no signs of fatigue, normal temperature)

Mild (increased temperature, scratching in the throat)

Moderate (fever, muscle aches, increased tiredness, and cough)

Severe (hospitalization)

R: Did any of the demographics such as age, gender, etc.… play a moderating role in the relationships explored in the study?

A: As shown by the Random Forest machine-learning algorithm for COVID-19 age and gender had only miniscule power to discriminate between the two classes of subjects. But thank you, we added the supplementary table (Supplement 1) with the risk ratios of upper respiratory tract infections for male and female.

R: Did the authors collect information on the type and intensity of activity performed by the physically active group? If yes, please report it.

A: Unfortunately we did not. We noted the need of type, intensity and length of physical exercise for future assessments at the end of discussion. 

R: It seems an important recent reference in this area is missing: “Sallis R, et al. Br J Sports Med 2021;0:1–8. doi:10.1136/bjsports-2021-104080”. This study concludes that those meeting PA guidelines consistently reported less severe COVID symptoms compared to those who were consistently inactive. I suggest adding this reference to the introduction to be able to make a stronger case for the current study’s hypothesis and the results. 

A: Thank you for that. We added this reference to the introduction.

Reviewer 4 Report

The study by Bielik V et al explored the incidence and course of COVID-19 and the risk of an upper respiratory tract infection in a group of people with physically active lifestyles. The authors reported  that cold-water swimming and physical activity did not decrease the risk of COVID-19 in recreational athletes. However, a physically active lifestyle might have a positive effect on the rate of incidence of acute respiratory infection and on the severity of COVID-19 symptoms.

A few major issues with the manuscript:

  1. It seems the data in Table 1 do not match what was described in the text.
  2. It is unclear how predictors were selected for the ROC curve
  3. The discussion were disorganised and unrelated to the data of the study. The authors discussed mitochondrial homeostasis in paragraph 2, which was not related to any of the data in the manuscript. The authors then discussed CRF and METs in paragraph 3, which were not assessed in this study. The authors then started to talk about brown adipose tissue and muscle damage, which were not measured in this study.

Author Response

R: 1. It seems the data in Table 1 do not match what was described in the text.

A: Thank you for this comment. Text was corrected. Table 1 was renamed Table 2 and new table (Table 1) with information of cohort was added.

R: 2. It is unclear how predictors were selected for the ROC curve.

A: Thank you for this comment. We added it to manuscript. All the studied predictors (frequency of inflammation of upper respiratory tract, weight, height, age, gender etc.) were fed into RandomForest. RF ranked them by variable importance. No formal feature selection was performed. 

R: The discussion were disorganised and unrelated to the data of the study.

A: We thank the reviewer for his comment. In the discussion we present a speculative model of the protective role of physical activity against covid-19. In the following are the reasons, which in our opinion support the structure and organization of the discussion.  Certain improvements, reorganization, and the information on limitations of the study were implemented:

  1. Mitochondrial homeostasis – We believe that mitochondrial fitness could explain why subjects with higher cardio-respiratory fitness may have lowered risk of severe COVID-19. Certainly, more data and research are needed to make the story round. As suggested by Nunn et al. 2020: “Unlike most sedentary modern humans, one of the natural hosts for the virus, the bat, has to “exercise” regularly to find food, which continually provides a powerful adaptive stimulus to maintain functional muscle and mitochondria. In effect the bat is exposed to regular hermetic stimuli, which could provide clues on how to resist this virus.” 
  2. CRF and METs. The limitation of this study is that the measure of physical activity was self-reported and there was no measure of the intensity. However, we assumed that those who reported regular physical activity and sports several times a week have better cardiorespiratory fitness than those who do not meet the recommendations. 
  3.  Adipose tissue browning is being discussed because of its relationship with the cold-water swimming and health benefits. Therefore, novice swimmers were excluded, as we were not certain whether after few cold-water baths the adipose tissue browning might be induced.

Round 2

Reviewer 4 Report

Thanks the authors for addressing the comments!

Please correct some typos and grammar in the manuscript.

Author Response

Thank you again for detailed control. We corrected typos and grammar in the manuscript and professional scientific english editor and native speaker performed final check .